# Bone mineral density among children living with HIV failing first-line anti-retroviral therapy in Uganda: A sub-study of the CHAPAS-4 trial

Eva Natukunda[1]*, Alex Szubert[2], Caroline Otike[1], Imerida Namyalo[1], Esther Nambi[1], Alasdair Bamford[2], Katja Doerholt[2], Diana M. Gibb[2], Victor Musiime[1,3], Phillipa Musoke[3,4]

1 Joint Clinical Research Centre, Kampala, Uganda, 2 Medical Research Council Clinical Trials Unit at University College London, London, United Kingdom, 3 Department of Paediatrics, College of Health Sciences, Makerere University Kampala, Kampala, Uganda, 4 Makerere University-Johns Hopkins University Research Collaboration (MUJHU CARE), Kampala, Uganda

* enatukunda@jcrc.org.ug

**Data Availability Statement:** The data underlying the results presented in the study are available from the institution (jcrc@jcrc.org.ug). Per the

## Abstract

### Background

Children living with perinatally acquired HIV (CLWH) survive into adulthood on antiretroviral therapy (ART). HIV, ART, and malnutrition can all lead to low bone mineral density (BMD). Few studies have described bone health among CLWH in Sub-Saharan Africa. We determined the prevalence and factors associated with low BMD among CLWH switching to second-line ART in the CHAPAS-4 trial (ISRCTN22964075) in Uganda.

### Methods

BMD was determined using dual-energy X-ray Absorptiometry (DXA). BMD Z-scores were adjusted for age, sex, height and race. Demographic characteristics were summarized using median interquartile range (IQR) for continuous variables and proportions for categorical variables. Logistic regression was used to determine the associations between each variable and low BMD.

### Results

A total of 159 children were enrolled (50% male) with median age (IQR) 10 (7–12) years, median duration of first-line ART 5.2(3.3–6.8) years; CD4 count 774 (528–1083) cells/mm$^3$, weight—for–age Z-score -1.36 (-2.19, -0.65) and body mass index Z-score (BMIZ) -1.31 (-2.06, -0.6). Low (Z-score$\leq$ -2) total body less head (TBLH) BMD was observed in 28 (18%) children, 21(13%) had low lumbar spine (LS) BMD, and15 (9%) had both. Low TBLH BMD was associated with increasing age (adjusted odds ratio [aOR] 1.37; 95% CI: 1.13–1.65, p = 0.001), female sex (aOR: 3.8; 95% CL: 1.31–10.81, p = 0.014), low BMI (aOR 0.36:95% CI: 0.21–0.61, p<0.001), and first-line zidovudine exposure (aOR: 3.68; 95% CI: 1.25–10.8, p = 0.018). CD4 count, viral load and first- line ART duration were not associated

Joint clinical research centre data policy, the data is provided on request from the institution.

**Funding:** The funding was obtained from EDCTP, Gilead Sciences and Janssen Pharmaceuticals and was received by Diana M Gibb. The funds for conference attendance were provided by Viatris and received by Diana M Gibb.

**Competing interests:** The authors have declared that no competing interests exist.

with TBLH BMD. Low LS BMD was associated with increasing age (aOR 1.42; 95% CI: 1.16–1.74, p = 0.001) and female sex: (aOR 3.41; 95% CI: 1.18–9.8, p = 0.023).

## Conclusion

Nearly 20% CLWH failing first-line ART had low BMD which was associated with female sex, older age, first-line ZDV exposure, and low BMI. Prevention, monitoring, and implications following transition to adult care should be prioritized to identify poor bone health in HIV+adolescents entering adulthood.

## Introduction

Globally, 38.4 million people are living with HIV (PLWH), with 1.7 million children aged 0–14 years. There were 160,000 new paediatric infections in 2021 with 90% living in Sub-Saharan Africa. Among children living with HIV (CLWH), 52% have access to antiretroviral therapy (ART) [1]. The World Health Organization (WHO) recommends that ART should be is initiated as soon as the HIV diagnosis is made in infants, children and young people [2]. Children now live longer into adulthood because of early ART initiation hence, the relative impact of long-term HIV and ART- related comorbidities, such as low bone mineral density (BMD) is becoming much more relevant. Low childhood BMD may result in suboptimal peak bone mass in adulthood and may predispose to osteoporosis and fracture risk in adulthood [3].

Low BMD has been previously described in PLWH and CLWH but most studies have been conducted in high-income countries [4, 5]. The prevalence of low BMD ranges from 20% to 80% in adults living with HIV [6, 7], and 4–32% in CLWH [8–10]. Few studies have been conducted in resource-limited settings where the majority of CLWH reside and where malnutrition, which also contributes to low BMD, is common [8, 10–12].

The aetiology of low BMD in the context of HIV infection is multifactorial. Relevant factors include uncontrolled HIV with a high viral load which favors osteoclast recruitment, low Vitamin D levels, low body mass index (BMI) and physical inactivity [13, 14]. Some antiretroviral drugs, including protease inhibitors, tenofovir disoproxil fumarate (TDF), and zidovudine (ZDV) contribute to low BMD while nevirapine (NVP) may be protective [8, 13]. Initiation of ART in treatment naïve adults or switching treatment due to virologic failure has also been found to be associated with reduced BMD in the first year followed by stabilization [10, 11, 13].

Bone health has not been widely studied in resource-limited settings yet it has the potential to be a long-term complication in PLWH. Under recognition is in part due to limited access to diagnostic dual-energy X-ray absorptiometry (DXA) facilities that detect low BMD before fracture occurrence.

In this sub-study of the CHAPAS-4 trial, we describe the baseline characteristics of a cohort of patients enrolled in the toxicity sub-study of the CHAPAS 4 trial in addition to reporting the bone health status of children switching to second-line ART at enrolment. We also examined the prevalence and risk factors associated with low BMD among children switching to second-line ART.

## Materials and methods

We conducted a cross-sectional study using baseline data from the toxicity sub-study of the CHAPAS 4 clinical trial, an ongoing open-label trial evaluating the virologic response to alternative second line antiretroviral therapy in CLWH.

In CHAPAS 4, 919 children aged 3–15 years failing on their 2NRTI+NNRTI- based first-line regimen with a viral load of 400copies/ml or more were randomized to receive tenofovir alafenamide/emtricitabine (TAF/FTC) or standard of care (abacavir/lamivudine [ABC/3TC] or zidovudine/lamivudine [ZDV/3TC], depending on the first-line treatment. Children were also randomized in a factorial design to dolutegravir (DTG) or darunavir/ritonavir (DRV/r) or atazanavir/ritonavir (ATV/r) or lopinavir/ritonavir (LPV/r). They are being followed for 96 weeks. In the toxicity sub-study, 444 children from study sites in Zimbabwe and Uganda were simultaneously enrolled to evaluate the potential renal and bone effects of TAF. Of these, 159 participants aged 5 years and above were enrolled at Joint Clinical Research Centre (JCRC) and consented to a DXA scan. The Centre recruited study participants from the paediatric clinic in the facility and neighbouring public health facilities in Wakiso, Mpigi, and Mukono districts. Reference BMD Z-scores were available for children who were ≥ 5 years and hence children who were 5 years and above were included for this study [15].

## Measurements

Data were collected at enrolment into the CHAPAS 4 trial from January 2019 to March 2021. A case report form was used to collect sociodemographic and clinical data. These included sex, age, and WHO classification of HIV disease [16], first-line ART and duration on treatment. Weight was measured using a Seca®weight scale, height was measured by wall mounted Seca® 206 stadiometer. BMI was calculated as weight (kg)/height (m$^2$). Weight, height and BMI Z-scores were measured using the British 1990 reference data [17]. Blood samples were collected to determine the viral load and CD4 cell counts by real-time HIV-RNA (COBAS Ampliprep/Taqman 96 analyzer-detection range of 20–10,000,000 copies per ml) and BD FACS Calibur respectively.

BMD was measured using one DXA scan (Hologic Discovery Wi Apex 13.5, Hologic Bedford, MA USA) for all participants. DXA scans were conducted by an experienced technician while the participant was lying still and flat on an examination table. A posterior–anterior scan was obtained for four lumbar spines (L1-L4) and the whole body in the array mode. The lumbar spine (LS) BMD assesses trabecular bone density while the whole- body is largely cortical. The head was excluded from the analysis as recommended by International Society for Clinical Densitometry 2019 (ISCD) [18]. BMD was measured in grams per square centimetre. Normative values for sex, age, race and height adjusted Z-scores were obtained from bone mineral density children study (BMDCS) reference data [15]. Low BMD was defined as a LS or total body less head (TBLH) BMD Z-score of less than or equal to -2 according to the ISCD [18].

## Quality control

A standard operating procedure was developed for participant preparation for BMD determination. The DXA scanner was calibrated by the technician daily for accuracy of the spine BMD determination with a phantom scan, standardization of the whole-body BMD determination was performed annually and a uniformity test was performed weekly.

## Ethical review

The main CHAPAS 4 trial was approved by the JCRC institutional review board (reference–JC 1417).

Caretakers of the enrolled children gave written informed consent; children aged eight years and above gave written informed assent as appropriate and according to the knowledge of their HIV status.

## Statistical analysis

Data were exported to STATA (StataCorp Texas USA) for analysis. Participant characteristics were summarized using median with interquartile range (IQR) for numerical data. Categorical data were summarized using frequencies and proportions. Low BMD was defined as either a height adjusted TBLH BMD or LS BMD z-score less than or equal to -2.

To measure the association of low BMD with other exposure variables, a logistic regression model was used. Bivariate analysis was performed for each of the independent variables to determine whether they were independently associated with low BMD using odds ratios and p-values. In multivariate analysis, stepwise backward elimination method was used to identify the independent variables with P-values <0.05 which were considered statistically significant. Confounding was assessed for all the non-significant independent variables and a relative difference of 10% or more between the crude and adjusted OR was considered as confounding.

## Results

A total of 159 CLWH failing their first-line ART regimen were enrolled. The median age was 10.0 years (IQR 7.0, 12.0) and 79/159 (49%) participants were male. The median BMI z-score was -1.31 (IQR-2.06, -0.60). The median age at ART initiation and duration of first-line ART were 4.0 (IQR 2.0, 7.0) and 5.2 (IQR 3.3, 6.8) years, respectively. The median CD4 count was 774 (528, 1083) cells/ul and all children had viral loads ≥400 copies/ml, as per the main trial inclusion criteria. The majority of the participants were stable: 144 (90.6%) had WHO clinical stage 1 and 2, with only 15 (9.45%) ever having stage 3 and 4 (**Table 1**).

Associations with low TBLH BMD are summarised in **Table 2**. In the bivariate analysis, participants exposed to ZDV-based first-line therapy were more likely to have low TBLH BMD compared to those on ABC crude OR [cOR] = 3.2, p = 0.012). Increasing age (cOR = 1.33 per year older p < 0.001) and BMI (cOR = 0.43 per unit increase, p<0.001) were associated with low TBLH BMD. There was no significant association between low TBLH BMD and absolute CD4 count, viral load, time on first- line ART or height.

Associations with low LS BMD at bivariate analysis are presented in **Table 3**. Females were more likely to have low LS BMD compared to males (cOR = 2.8, p = 0.04). Older age was also associated with low LS BMD (cOR = 1.38 per year older p = 0.001), as was older age at first line ART initiation (cOR = 1.17, p = 0.037).

The results of multivariate analysis are summarized in **Table 4**.

In the multivariate analysis of TBLH BMD, first-line ZDV exposure (aOR = 3.68, p = 0.018), older age (aOR = 1.37 per year older, p = 0.001), female sex (aOR = 3.8, p = 0.014) and lower BMI (aOR = 0.36 per unit higher, p<0.001) were independently associated with low TBLH BMD.

Females were more likely to have a lower LS BMD compared to males (aOR = 3.41, p = 0.023), and older age (aOR = 1.42 per year older, p<0.001) was independently associated with low LS BMD.

## Discussion

The overall prevalence of low BMD in our study was high at 21%, with higher detection by TBLH (18%) compared to the LS (13%), whereas 9% children had both. The factors associated with low TBLH BMD were age, sex and low BMI, which is consistent with the results of other studies. In our study, ZDV exposure was the only drug-related factor associated with low BMD. However, since this was a cross-sectional study, associations do not imply causality.

The prevalence of low BMD in this study is comparable to that in other studies that have described a high prevalence of low BMD in CLWH [8, 19]. However, a prevalence rate as low

**Table 1. Characteristics of study participants, N = 159.**

| Characteristic | Measure |
|---|---|
| Age: median (IQR) | 10 (7,12) |
| Male sex (n/%) | 79 (49.7) |
| **Anthropometry** | |
| Weight-for-age-Z-score (WAZ): median (IQR) | -1.36 (-2.19, -0.65) |
| Height-for-age-Z-score (HAZ): median (IQR) | -1.12 (-1.58, -0.10) |
| BMI- for- age-Z-score: median (IQR) | -1.31 (-2.06, -0.60) |
| BMI- for- age-Z-score:<-2 (%) | 18 (11.3) |
| **HIV characteristics** | |
| CD4 count (cells/ul): median (IQR) | 774 (528, 1083) |
| CD4%, median (IQR) | 31 (22, 37) |
| Viral load ($\log_{10}$copies/ml): median (IQR) | 4.25 (3.73, 4.85) |
| **WHO Stage (n/N)** | |
| Stage 1&2 | 144 (90.6%) |
| Stage 3&4 | 15 (9.4%) |
| **First- line ART regimen (n/N)** | |
| ABC- based | 75 (47.2%) |
| ZDV- based | 83 (52.2%) |
| EFV- based | 80 (50.3%) |
| NVP- based | 79 (49.7%) |
| Age at First-line ART initiation: median (IQR) | 4 (2,7) |
| **First- line ART duration in years**: median (IQR) | 5.21 (3.33, 6.84) |
| **Bone mineral density measures** | |
| **Lumbar spine** | |
| BMD g/cm$^3$: median (IQR) | 0.545 (0.487–0.648) |
| LSBMD$_{HAZ:}$ median (IQR) | -0.99 (-1.46, -0.34) |
| Low lumbar BMD n/N (%) | 21 (13.2) |
| **TBLH** | |
| BMDg/cm$^3$: median (IQR) | 0.656 (0.577, 0.732) |
| TBLH $_{HAZ:}$ median (IQR) | -1.37 (-1.87, -0.81) |
| Low TBLH BMD n/N (%) | 28 (17.6) |
| Low LSBMD with low TBLH BMD | 15 (9.4) |

Abbreviations: IQR = interquartile range, BMI = body mass index, LSBMD = lumbar spine bone mineral density, TBLH BMD = total body less head bone mineral density, HAZ = height adjusted Z- score; ZDV, zidovudine; NVP = nevirapine; EFV = efavirenz, ABC = abacavir

as 4–7% has been reported in well-resourced countries [20–22]. Associated inadequate nutrition may have contributed to the higher prevalence of low BMD, given that our study population resides in a resource-limited setting, where balanced diets are not readily available [23]. In addition, half of the participants had prolonged exposure to efavirenz (EFV), which disrupts calcium and vitamin D metabolism, that are essential for bone development [24]. Low vitamin D status has been described among CLWH in Africa with a prevalence of 17.3% [25]. Although it was not assessed in this study, a recent study among Ugandan CLWH found an association between EFV use and vitamin D deficiency [26].

Increasing age was independently associated with low BMD, which may be explained by the delayed puberty that has been observed in CLWH and is associated with delayed skeletal growth [19, 27]. More females in this study had low BMD compared to males in both TBLH

**Table 2. Bivariate total body less head BMD analysis, N = 159.**

| Characteristic | Abnormal BMD | Normal BMD | cOR | 95%CI | P-Value |
|---|---|---|---|---|---|
|  | N (%) | N (%) |  |  |  |
| **Sex** |  |  |  |  |  |
| Male | 10 (12.7) | 69 (87.3) | 1 |  |  |
| Female | 18 (22.5) | 62 (77.5) | 2.00 | 0.86–4.7 | 0.107 |
| **WHO stage** |  |  |  |  |  |
| Stage 1&2 | 24 (16.7) | 120 (83.3) | 1 |  |  |
| Stage 3&4 | 4 (26.7) | 11 (73.3) | 1.82 | 0.53–6.19 | 0.339 |
| **Regimen** |  |  |  |  |  |
| ABC- based | 7 (9.3) | 68 (90.7) | 1 |  |  |
| ZDV- based | 21 (25.0) | 63 (75.0) | 3.2 | 1.29–8.14 | 0.012 |
| EFV- based | 14 (17.5) | 66 (82.5) | 1 |  |  |
| NVP- based | 14 (17.7) | 65 (82.3) | 1.01 | 0.45–2.29 | 0.97 |
|  | Median (IQR) | Median (IQR) |  |  |  |
| Age | 12 (10,14) | 9 (7,12) | 1.33 | 1.13–1.57 | 0.001 |
| BMI-Z | -2.0 (-3.0, -1.0) | -1.0 (-2.0, -1.0) | 0.43 | 0.28–0.66 | <0.001 |
| Abs CD4 count | 738 (564.5,1052) | 774 (524,1083) | 1.00 | 0.99–1.00 | 0.953 |
| CD4% | 31 (23,35.5) | 30 (22,37) | 1.00 | 0.96–1.03 | 0.948 |
| HIV RNA log $_{10}$ copies per ml | 4.33 (3.87,4.94) | 4.24 (3.73,4.84) | 1.05 | 0.62–1.78 | 0.86 |
| Time on first-line ART | 6.0 (4.0,7.0) | 5.0 (3.0,7.0) | 1.06 | 0.91–1.25 | 0.439 |
| Age at first-line ART start | 6.0 (4.0,8.0) | 4.0 (2.0,6.0) | 1.22 | 1.06–1.399 | 0.05 |

Abbreviations: IQR = interquartile range, BMI = body mass index, LSBMD = lumbar spine bone mineral density, TBLH BMD = total body less head bone mineral density, HAZ = height adjusted Z- score; ZDV = zidovudine, EFV = efavirenz, NVP = nevirapine, ABC = abacavir, ART = antiretroviral therapy, BMD = bone mineral density (BMD).

and LS which is consistent with other studies [8]. Sex differences in pre-pubertal BMD are conflicting. However, females tend to have smaller bones compared to males. BMD is a two-dimensional measurement that tends to underestimate the BMD for smaller bones [28]. The females in the study population may have smaller bones compared to the reference population that had height- for- age Z-scores ranging from 0.2 to 0.45 [29].

Low BMD was associated with low BMI in this study similar to the results of other studies where adequate BMI- for- age was correlated with increased bone mass. Bones adapt to mechanical loading and an increase in load results in strengthening of bone by remodeling itself to accommodate the increased load, while a decrease in load causes bone weakening [6, 30, 31].

Data regarding the effect of ART on bone are variable. In previous studies, use of ZDV or EFV was associated with low BMD, which is consistent with our findings where low BMD was associated with ZDV use. Kim et al found bone loss in Korean adults who were exposed to ZDV for over a year [32]. Similarly in a longitudinal study, patients who switched to ZDV had a small but significant decrease in BMD compared to those taking ABC who remained stable [33]. This is also consistent with the in vitro observation that ZDV stimulates osteoclast activity in mice [34]. Paediatric data regarding ZDV and bone density is limited, although Bunders et al found no association between ZDV and low spinal BMD. Similarly, there was no association of ZDV with spinal BMD in our study [20]. There was also no significant association between EFV and BMD in our study. Adult studies have associated EFV with reduced BMD but there is limited data regarding BMD and EFV in the paediatric population [35]. However there was bone accrual in South African children who switched from lopinavir/ritonavir based

**Table 3. Bivariate Lumbar spine bone mineral density analysis, N = 159.**

| Characteristics | Abnormal | Normal | cOR | 95%CI | P-Value |
|---|---|---|---|---|---|
| | N (%) | N (%) | | | |
| **Sex** | | | | | |
| Male | 6 (7.59) | 73 (92.41) | | | |
| Female | 15 (18.75) | 65 (81.25) | 2.8 | 1.03–7.67 | 0.04 |
| **WHO stage** | | | | | |
| Stage 1&2 | 19 (13.19) | 125 (86.81) | 1 | | |
| Stage 3&4 | 2 (13.3) | 13 (86.67) | 1.01 | 0.21–4.84 | 0.988 |
| **Regimen** | | | | | |
| ABC- based | 7 (9.33) | 68 (90.7) | 1 | | |
| ZDV- based | 21 (25) | 63 (75) | 1.94 | 0.74–5.11 | 0.18 |
| EFV- based | 11 (13.75) | 69 (86.25) | 1 | | |
| NVP- based | 10(12.7) | 69(87.34) | 0.91 | 0.36–2.27 | 0.84 |
| | Median (IQR) | Median (IQR) | | | |
| Age | 13 (11,14) | 9.5 (7.0,12.0) | 1.38 | 1.14–1.68 | 0.001 |
| BMIZ | -2.0 (-2.0, -1.0) | -1.0 (-2.0, -1.0) | 0.80 | 0.54–1.19 | 0.277 |
| Weightz | -2.0 (2.0) | -1.0 (2.0) | 1.09 | 0.78–1.55 | 0.591 |
| Abs CD4 | 808 (615,970) | 770 (505,1083) | 1.00 | 0.99–1.00 | 0.782 |
| CD4% | 32 (26,37) | 30 (22,37) | 1.025 | 0.98–1.07 | 0.251 |
| HIV RNA | 3.97(3.49,4.54) | 4.54(3.85,4.886) | 0.56 | 0.30–1.05 | 0.07 |
| Time on first-line ART | 6.0 (5.0,8.0) | 5.0 (3.0,7.0) | 1.149 | 0.96–1.37 | 0.119 |
| Age at first-line ART start | 6.0 (4.0,8.0) | 4.0 (2.0,6.0) | 1.17 | 1.01–1.37 | 0.037 |

Abbreviations: IQR = interquartile range, BMIZ = body mass index Z-score, Weightz = weight Z score, HIV RNA = HIV viral load, ART = anti-retroviral therapy, EFV = efavirenz, NVP = nevirapine, ABC = abacavir, ZDV = zidovudine

ART to EFV [36]. This finding may have been due to change to a drug with a better bone safety profile but is difficult to compare with our study findings. Bunders et al found no association between low BMD and EFV in children which is similar to our findings [13, 19, 29, 30, 32, 33]. The mechanism for no association is unclear but may be due to the difference in the pharmaco-kinetics of EFV in children that are distinct from adults. Plasma clearance of EFV in children is faster than in adults and hence may have a less significant effect on bone health [37, 38].

The mechanisms through which HIV causes low BMD include osteoclast activation by the HIV viral protein gp120 and production of pro-inflammatory cytokine TNFα and interleukin

**Table 4. Multivariate analysis of baseline characteristics and bone mineral density, N = 159.**

| TBLH BMD | | | |
|---|---|---|---|
| Characteristic | aOR | 95%CI | P-Value |
| Age (/year older) | 1.37 | 1.13–1.65 | 0.001 |
| Female sex | 3.80 | 1.31–10.81 | 0.014 |
| ZDV compared to ABC based regimen | 3.68 | 1.2510.80 | 0.018 |
| BMIZ (/unit increase) | 0.36 | 0.21–0.61 | <0.001 |
| **LS BMD** | | | |
| Age (/year older) | 1.42 | 1.16–1.74 | 0.001 |
| Female Sex | 3.41 | 1.18–9.8 | 0.023 |

Abbreviations: aOR = adjusted odds ratio, BMIZ = body mass index Z-score, LS BMD = lumbar spine bone mineral density, TBLH BMD = total body less head bone mineral density, ZDV = zidovudine, ABC = abacavir

6 that increase bone resorption. There was no association between HIV viral load and BMD in our study. Viral load at a single time point may not be an accurate measure of lifetime HIV virological control. Different studies have reported inconsistent results. Dimeglio et al did not adjust for height hence they found that a high viral load was associated with low LS BMD. However, other paediatric studies found no association after height adjustment for BMD was done similar in our study where [21, 39, 40] height adjustment for BMD Z- score was performed.

There are several potential limitations of this study. The DXA scanner does not measure volumetric density but only measures areal density, therefore underestimates the BMD for smaller bones.

There are no local or African BMD reference data for children. The reference data used for determining BMD Z scores were obtained from the BMDCS that used a healthy black American population. It is possible that we may have overestimated low BMD due to the geographical, environmental and dietary differences between our study population and the reference data [15]. Thirdly there was no HIV negative control group from a similar environment. This was a single-site study which limits the generalizability.

## Conclusion

Low BMD is prevalent among Ugandan CLWH with virologic failure following first-line ART. Low TBLH BMD was associated with older age, female sex, low BMI and first- line ZDV exposure.

Health intervention programs should promote improved nutrition and also minimize the use of regimens associated with low BMD to prevent potential future fracture risk in low-income settings. Follow-up of these children in the CHAPAS-4 trial to a minimum of 96 weeks will enable us to evaluate the long-term effects of second-line ART on BMD. Future research should focus on therapeutic and preventive interventions and on monitoring the impact of these findings on bone health following transition to adult care.

## Acknowledgments

This work was done as part of the CHAPAS 4 trial. The authors would like to thank the children and their caretakers who volunteered to participate in this study.

We thank the data collection team at the Joint Clinical Research Centre Kampala.

We thank Gilead sciences, CIPLA, ViiV healthcare and Janssen for providing the trial drugs.

## Author Contributions

**Conceptualization:** Eva Natukunda, Diana M. Gibb.

**Data curation:** Eva Natukunda, Alex Szubert, Caroline Otike.

**Formal analysis:** Eva Natukunda, Alex Szubert, Caroline Otike.

**Funding acquisition:** Diana M. Gibb.

**Investigation:** Eva Natukunda, Imerida Namyalo, Esther Nambi.

**Methodology:** Eva Natukunda.

**Project administration:** Eva Natukunda, Esther Nambi.

**Resources:** Diana M. Gibb.

**Supervision:** Victor Musiime, Phillipa Musoke.

**Validation:** Eva Natukunda.

**Visualization:** Eva Natukunda.

**Writing – original draft:** Eva Natukunda.

**Writing – review & editing:** Eva Natukunda, Alex Szubert, Caroline Otike, Esther Nambi, Alasdair Bamford, Katja Doerholt, Diana M. Gibb, Victor Musiime, Phillipa Musoke.

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
