## [Decision Letter · Decision Letter 0]

10 Apr 2023

PONE-D-23-05357Bone mineral density among children living with HIV failing first-line antiretroviral therapy in Uganda. A sub-study of the CHAPAS-4 trial.PLOS ONE

Dear Dr. Natukunda,

Thank you for submitting your manuscript to PLOS ONE. After careful consideration, we feel that it has merit but does not fully meet PLOS ONE’s publication criteria as it currently stands. Therefore, we invite you to submit a revised version of the manuscript that addresses the points raised during the review process.

The reviewers have made comments and edits to the manuscript. Kindly consider all of these and respond as appropriate.

We look forward to receiving your revised manuscript.

Kind regards,

Chika Kingsley Onwuamah, Ph.D.

Academic Editor

PLOS ONE

Journal Requirements:

"This work was done as part of the CHAPAS 4 trial. The authors would like to thank the children and their caretakers who volunteered to participate in this study.

We thank all the data collection team at the Joint Clinical Research Centre Kampala

CHAPAS 4 trial is mainly funded by the European and Developing Countries Clinical Trials Partnership (grant number TRIA2015-1078) and Medical Research Council (MRC)

Additional support was received from: Gilead Sciences, Janssen Pharmaceuticals, ViiV Health care and CIPLA"

"DMG received the award from European and Developing Countries Clinical Trials Partnership (EDCTP) grant number TRIA2015-1078.https://www.edctp.org

Additional support was provided by Gilead sciences , https://www.gilead.com,Jannsen Pharmaceuticals,https://www.janssen.comViiV Health care,https://viivhealthcare.com

The sponsors had no role in study desin,data collection and analysis,decision to publish or preparation of the manuscript"

Reviewers' comments:

Reviewer's Responses to Questions

**Comments to the Author**

1. Is the manuscript technically sound, and do the data support the conclusions?

Reviewer #1: Partly

Reviewer #2: Yes

2. Has the statistical analysis been performed appropriately and rigorously? 

Reviewer #1: No

Reviewer #2: Yes

3. Have the authors made all data underlying the findings in their manuscript fully available?

Reviewer #1: Yes

Reviewer #2: Yes

4. Is the manuscript presented in an intelligible fashion and written in standard English?

Reviewer #1: Yes

Reviewer #2: Yes

5. Review Comments to the Author

Reviewer #1: This is an interesting study that highlights the prevalence of co-morbidities such as low BMD in CLWH, especially pertaining to the cohort that is switching treatment due to virologic failure.

These observations are even more relevant in populations that are resource-limited and do not have access to a balanced diet.

The manuscript if re-structured in the context of the larger question addressed in the CHAPAS4 trial will have significant implications.

1. Although this study does an excellent job characterizing the BMD, LSBMD, and TBLH in CLWH, it is unclear how these findings address the questions asked in the CHAPAS4 trial and specifically pertaining to the toxicity sub-study.

2. The authors have not been able to tease out the impact of the high viral load and the first-line ART (BAC-based, ZDV based, etc.) on BMD.

3. Did the authors normalize or control for low BMI as a confounder to low BMD, since the study was conducted in a low-resource environment?

4. It is unclear from the results and the lack of description in the manuscript, the characteristics (if any) of the age-matched controls or if the "normal" BMD is that of controls.

5. There should be 2 separate control cohorts (if feasible) included in this study

a. Uninfected controls with normal BMD

b. CLWH that responded to the first line of ART

6. There is no graphical representation of the data.

7. Details of the DXA and parameters of the scans and the analyses need to be included. Was the same DXA machine used for performing scans on all participants?

8. The interquartile range is a useful measure of statistical dispersion because it is less sensitive to outliers. It will be useful to compare the median to that of control subjects to identify any potential outliers.

Reviewer #2: Thank you for a well written paper on the prevalence and associated factors of low BMD in your study population of children switching to second line ART due to virological failure. The findings showed a high prevalence of low BMD and several associated factors, including age, sex, BMI and AZT use. I think highlighting the need for following up these children/adolescents as they transition into adult care and assessing the long term outcomes, in terms of possible fracture risk, is very important as this is not a risk that is frequently monitored/assessed in the resource-limited settings where these adolescents are being cared for. I recommend that the article is accepted for publication after minor revisions.

Major Revisions

Nil

Minor Revisions

Abstract Results: Please review minor grammar, capitals and spaces before/after brackets as needed

Abstract Conclusion: “and development of cost-effective screening tools to identify poor bone health in HIV+ adolescents entering adulthood.” Development of screening tools is not mentioned in your main text. Either add this into your main text (because it is very important) or adjust your abstract conclusion to reflect your main text conclusion

Introduction: “Because of early ART initiation…” Suggest rephrasing this sentence so that it does not begin with “Because”

Introduction: “ART-related comorbidities including low bone mineral density (BMD) are becoming…” Suggest adding commas to read “comorbidities, including low bone mineral density (BMD), …”

Introduction: “where the majority of children” Should this read “where the majority of CLWH reside”

Introduction: “and where malnutrition that also contributes to low BMD is common” Suggest adding commas to read “and where malnutrition, which also contributes to low BMD, is common”

Introduction: “also contributes to low BMD is common. [8, 10-12] However BMD

is higher in the black population compared” The link between these two sentences needs to be more clearly explained

Introduction: “zidovudine (ZDV), contribute to” Suggest removing comma here

Introduction “also been documented to be associated” Suggest removing “as documented” from this sentence to make it easier to read

Introduction: “BMD is determined by z-scores in adolescents and children according to the International Society for Clinical Densitometry 2019(ISCD).[16] Low BMD is defined as z--scores less or equal to -2 defines low BMD.[16]. Suggest moving this paragraph to methods. The second sentence of this paragraph has repetition and needs to be reviewed.

Methods: “lamivudine(ZDV/3TC), depending on first-line). Children” Suggest removing bracket at the end of the sentence.

Methods: “Zimbabwe and Uganda were consecutively enrolled” Should this say simultaneously instead of consecutively?

Methods: “Reference BMD Z-scores were available for children who were ≥ 5 years.” If this is the reason only children > 5 were enrolled suggest making this more clear

Methods: “and classification of HIV disease[18], first” Please clarify what classification was used in the text

Methods: “The LS BMD assesses trabecular” This is the first time this abbreviation is used, please put lumbar spine (LS)

Methods: “children aged eight years and above gave written informed assent according to and as appropriate based on, knowledge of HIV status.” This sentence is not clear, please review (possibly remove the comma and say knowledge of their HIV status)

Methods: “or (LS) BMD z-score less than or equal to -2.” Remove brackets around LS

Results: “with only 15(9.45%) ever had stage 3 and 4.” Suggest changing to “ever having”

Results: “Associations with low total body less head (TBLH) BMD are summarised” Can just use TBLH here as abbreviation explained previously in the text. It will make it easier to read

Results: “Increasing age was associated with low TBLH BMD (cOR=1.33 per year older=0.001). Low BMI was also associated with low BMD (cOR=0.43 per unit increase. p<0.001).” Suggest making this into one sentence.

Results: “Associations with low lumbar spine (LS) BMD” Can just use LS here as abbreviation explained previously

Results: “LSBMD” separate into LS BMD throughout text

Results: “Multivariate analysis is summarized in Table 4,” Is this part of the next paragraph? Not clear

All tables: all abbreviations used in the table must be shown below the table, including ARVs. Please check this with all your tables

Discussion: “Paediatric data regarding ZDV and bone density is limited though in one study bunders et al found no association with spinal BMD.Similarly,there was no association with spinal BMD in our study [22]” Capitalise Bunders. Please clarify these two sentences…no association with low or worsening BMD?

Discussion: “HIV viral protein gp120 causes osteoclast activation while pro-inflammatory cytokine TNFα and interleukin 6 increase bone resorption. There was no association between HIV viral load and BMD in our study. Viral load at a single time point may not be an accurate measure of lifetime HIV virological control. Different studies reported inconsistent results. Dimeglio et al found high viral load was associated with low LSBMD while other paediatric studies found no association after height adjustment for BMD was done as in our study. In our study height adjustment for BMD Z- score was done.[23, 41, 42]

It would be useful to link the first sentence into the rest of the paragraph more, by making it clear that those are potential ways HIV causes low BMD. Did Dimeglio et al not adjust for height and that is why they potentially found a low LSBMD compared to other studies that did height adjust? This is not clear in this paragraph. Your references at the end should be after the second to last sentence where the studies are mentioned.

Discussion: “Thirdly there was no control group from a similar environment.” Suggest adding in what sort of control group may have improved the study…HIV negative children from the same environment? HIV positive children who are virally suppressed?

Discussion: “And this was a single- site study limiting generalizability.” Please rephrase so that the sentence does not begin with And.

Discussion: “Vitamin D is essential for bone formation. Low Vitamin D status has been described among children living in Africa with a prevalence of 17.3%.[43] We did not assess vitamin D status but our study participants were exposed to EFV that may interfere with vitamin D metabolism.” This is already mentioned in the second paragraph of your discussion. Suggest merging this paragraph into that second paragraph. It is a bit repetitive and out of place after the limitations.

6. PLOS authors have the option to publish the peer review history of their article (what does this mean?). If published, this will include your full peer review and any attached files.

Reviewer #1: No

Reviewer #2: No

---

## [Decision Letter · Decision Letter 1]

6 Jul 2023

Bone mineral density among children living with HIV failing first-line antiretroviral therapy in Uganda. A sub-study of the CHAPAS-4 trial.

PONE-D-23-05357R1

Dear Dr. Natukunda,

We’re pleased to inform you that your manuscript has been judged scientifically suitable for publication and will be formally accepted for publication once it meets all outstanding technical requirements.

Kind regards,

Chika Kingsley Onwuamah, Ph.D.

Academic Editor

PLOS ONE

Additional Editor Comments (optional):

Reviewers' comments:

Reviewer's Responses to Questions

**Comments to the Author**

1. If the authors have adequately addressed your comments raised in a previous round of review and you feel that this manuscript is now acceptable for publication, you may indicate that here to bypass the “Comments to the Author” section, enter your conflict of interest statement in the “Confidential to Editor” section, and submit your "Accept" recommendation.

Reviewer #1: All comments have been addressed

Reviewer #2: All comments have been addressed

2. Is the manuscript technically sound, and do the data support the conclusions?

Reviewer #1: Yes

Reviewer #2: Yes

3. Has the statistical analysis been performed appropriately and rigorously? 

Reviewer #1: Yes

Reviewer #2: Yes

4. Have the authors made all data underlying the findings in their manuscript fully available?

Reviewer #1: Yes

Reviewer #2: Yes

5. Is the manuscript presented in an intelligible fashion and written in standard English?

Reviewer #1: (No Response)

Reviewer #2: Yes

6. Review Comments to the Author

Reviewer #1: Please add the the parameters set in the statistical codes for variables that are potentially confounding in the results section. Refer to the author response to comment #3 made by the reviewer.

Reviewer #2: Thank you for your revisions made to the paper and I look forward to seeing the results from the follow up study that will assess the changes in BMD on second line regimens. I recommend that the article is accepted for publication. I have a few suggestions below. The article does not need further review if these changes are made.

Introduction line 1: Suggest removing the abbreviation PLWH from this sentence as in this sentence it reads “people are living with HIV” rather than people living with HIV. Suggest putting it into the first sentence of the second paragraph to read “Low BMD has been previously described in people living with HIV (PLWH) and

Introduction: “recommends that ART should be is initiated”: Remove “is”

Discussion: “Similarly, there was no association of ZDV with spinal BMD in our study” Suggest changing to read: “Similarly, there was no association of ZDV with low spinal BMD in our study”

Discussion: “height adjustment for BMD was done similar as in our study” Suggest changing to read “height adjustment for BMD was done, similar to our study

7. PLOS authors have the option to publish the peer review history of their article (what does this mean?). If published, this will include your full peer review and any attached files.

Reviewer #1: No

Reviewer #2: No

---

## [Editor Report · Acceptance letter]

12 Jul 2023

PONE-D-23-05357R1 

Bone mineral density among children living with HIV failing first-line anti-retroviral therapy in Uganda: A sub-study of the CHAPAS-4 trial. 

Dear Dr. Natukunda:

I'm pleased to inform you that your manuscript has been deemed suitable for publication in PLOS ONE. Congratulations! Your manuscript is now with our production department. 

Kind regards, 

on behalf of

Dr. Chika Kingsley Onwuamah 

Academic Editor

PLOS ONE